# Transcriptome Analysis Reveals Crosstalk between the Abscisic Acid and Jasmonic Acid Signaling Pathways in Rice-Mediated Defense against *Nilaparvata lugens*

**DOI:** 10.3390/ijms23116319

**Published:** 2022-06-05

**Authors:** Jitong Li, Lin Chen, Xu Ding, Wenyan Fan, Jinglan Liu

**Affiliations:** 1College of Horticulture and Plant Protection, Yangzhou University, Yangzhou 225009, China; dx120200130@yzu.edu.cn (J.L.); chenlin88@yzu.edu.cn (L.C.); dingxu115413@163.com (X.D.); fanwy0406@163.com (W.F.); 2Joint International Research Laboratory of Agriculture and Agri-Product Safety of the Ministry of Education, Yangzhou University, Yangzhou 225009, China

**Keywords:** brown planthopper, transcriptome analysis, insect resistance, abscisic acid (ABA), jasmonic acid (JA)

## Abstract

The brown planthopper (BPH) impacts both rice yield and quality. The exogenous application of abscisic acid (ABA) and jasmonic acid (JA) has been previously shown to induce rice resistance to BPH; however, the regulation of rice-mediated defense by these plant growth regulators is unclear. We applied exogenous JA and ABA to rice and analyzed molecular responses to BPH infestation. Nine RNA libraries were sequenced, and 6218 differentially expressed genes (DEGs) were generated and annotated. After ABA + BPH and JA + BPH treatments, 3491 and 2727 DEGs, respectively, were identified when compared with the control (BPH alone). GO enrichment and KEGG pathway analysis showed that the expression of several JA pathway genes (*OsAOS2,* encoding allene oxide synthase; *OsOPR,* 12-oxo-phytodienoic acid reductase; and *OsACOX,* acy1-CoA oxidase) were significantly up-regulated after ABA + BPH treatment. Furthermore, exogenous JA increased the expression of genes involved in ABA synthesis. Meanwhile, the expression levels of genes encoding WRKY transcription factors, myelocytomatosis protein 2 (*MYC2*) and basic leucine zippers (*bZIPs*) were up-regulated significantly, indicating that ABA and JA might function together to increase the expression of transcription factors during the rice defense response. The DEGs identified in this study provide vital insights into the synergism between ABA and JA and further contribute to the mechanistic basis of rice resistance to BPH.

## 1. Introduction

Rice (*Oryza sativa* L.) is consumed worldwide and feeds over half the global population. Due to incremental increases in the global population, rice yields need to be doubled over the next 40 years to meet incremental population increases [1]. Most rice cultivars show varying degrees of tolerance to abiotic and biotic stressors, which include drought, salinity, pathogens and pests. Frequent and widespread outbreaks of insect pests have caused extensive losses to growers and threaten food security [2,3], which has further emphasized the need to reduce the impact of pests on rice yield [4,5].

The brown planthopper (BPH), *Nilaparvata lugens* (Hemiptera: Delphacidae), is one of the most destructive pests in rice-growing areas of Asia and Southeast Asia [6]. BPH has a high growth rate and strong environmental adaptability. It directly feeds on the phloem of rice, lays eggs on the stem and transmits various plant viruses [7,8]. BPH has become a significant pest in China, and strategies for improving rice resistance to this pest are urgently needed [1,9].

Growth regulators play important roles in plant development and productivity [10] and also function in the plant defense response to pests and pathogens. Abscisic acid (ABA) and jasmonic acid (JA) are two important hormones that actively regulate plant defense, growth and development [11,12]. Treatment with exogenous ABA induced resistance to BPH and ultimately improved rice yield [13]. Further studies find that ABA-induced callose formation in rice is a mechanism for enhanced resistance to BPH [14]. In addition to its role in callose deposition, ABA can activate JA-mediated expression of disease resistance genes, and the interaction between ABA and JA can be synergistic [15,16,17,18] or antagonistic [19,20]. The ABA and JA signaling pathways further activate the expression of defense genes and increase the production of defense compounds that contribute to insect resistance [21,22].

We conducted this study to assess the defense response of rice to BPH after exogenous application of JA and ABA. In-depth transcriptome sequencing was conducted to identify potential candidate genes and their differential expression. Changes in the expression of transcription factors (TFs) involved in resistance and related genes were used to evaluate interactions between hormones, TFs and rice genes. Our findings provide evidence that JA and ABA cooperate to induce BPH resistance and provide valuable insights regarding insect resistance in rice.

## 2. Results

### 2.1. RNA-seq Assembly and Mapping

Transcriptome analysis was conducted on rice treated with BPH + ABA, BPH + JA and the control (BPH alone). Nine cDNA libraries were constructed; after removing unknown and low-quality reads, nearly 250 million reads were obtained. As shown in Table 1, the Q30 values of reads were higher than 90%. The Nipponbare genome in the RAP database (http://rapdb.dna.affrc.go.jp (accessed on 11 November 2021) was employed as a reference, and high-quality fragments were used for reference-guided transcriptome assembly with 79–93% of reads aligning with the reference genome (Table 1). The correlation between different samples is presented in Figure 1A.

Comparisons were made among the three treatments using fold-change ≥2 and false discovery rates (FDRs) ≤0.01 as standards. A total of 3491 genes were differentially expressed in the BPH and BPH + ABA treatments, with 1795 and 1696 exhibiting up- and down-regulation, respectively (Table 2). A total of 2727 differentially expressed genes (DEGs) were observed when comparing the BPH and BPH + JA treatments; 1674 genes were upregulated and 1053 were down-regulated. A total of 1223 genes were regulated between the two groups (BA and BJ); 519 genes were up-regulated, and 704 genes were down-regulated (Figure 1B).

### 2.2. Functional Classification, Annotation and Quality Assessment of DEGs

DEGs were compared with the GO (Gene Ontology), KEGG (Kyoto Encyclopedia of Genes and Genomes) and KOG (euKaryotic Ortholog Group) databases to obtain annotations and deduce potential functions. GO was used for the functional enrichment analysis of DEGs exposed to BPH, BPH + ABA and BPH + JA. GO classification results could be divided into the following: molecular function, cell composition and biological process. The top 30 terms enriched GO terms with the greatest significant differences were selected for further analysis. In the BPH + ABA treatment, the number of annotated genes categorized as a cell part, cell and organelle were 1138, 1134 and 1026, respectively. In terms of molecular function, catalytic ability (853) and binding activity (840) were the two most abundant categories (Figure 2A). The two most common types of biological processes were metabolic (1051) and cellular (1809) (Figure 2A). In the BPH + JA treatment, the number of annotated genes categorized as a cell part, cell and organelle were 1433, 1430 and 1276, respectively. Regarding molecular function, the predominant categories were catalytic ability and binding activity with 1226 and 1192 annotated genes, respectively. Two biological processes were well-represented, namely, metabolic (1457) and cellular (1118) (Figure 2B).

A total of 2700 genes were identified when the two groups (BJ and BA) were compared to the euKaryotic Ortholog Group (KOG) database using BLASTX (Figure 2C). As shown in Figure 2C, most genes were assigned to the R, T and O groups that represent ‘general function, prediction only’, ‘signal transduction mechanisms’ and ‘posttranslational modification, protein turnover, chaperones’, respectively.

### 2.3. Analysis of KEGG Metabolic Pathways

To ensure that the numbers of fragments accurately reflect transcript levels, the mapped reads and the length of transcripts in the samples were normalized. The fragments per kilobase of exon per million mapped fragments (FPKM) were used as an index to measure the expression level of transcripts, and the FPKM values of genes and transcripts were calculated. FDRs and *p*-values were calculated to measure the significance and the reliability of the three treatments in reducing bias. The DEGs selected by the three treatments had FDRs ≥0.1 and fold-change values >2 after eliminating biases.

The KEGG database was used for enrichment analysis, and the top 20 pathways with the highest enrichment significance were selected for further study. DEGs in the BPH vs. BPH + JA and BPH vs. BPH + ABA groups were compared (Figure 3). The horizontal axis shows the enrichment factor of DEGs in a pathway, while the vertical axis shows the name of the enrichment pathway. The top 20 enriched pathways were selected for further analysis. In the BPH + ABA group, a total of 778 DEGs were assigned to 27 KEGG pathways. Among these, the biosynthesis of flavonoids and flavonols had the highest significance. No significant changes were observed in the following categories: chlorophyll II metabolism; metabolic pathways; photosynthesis; ribosome; alanine, aspartate and glutamate metabolic pathway; β-alanine metabolism; or carbon fixation in photosynthesis (Figure 3A). In the BPH + JA treatment group, a total of 639 DEGs were assigned to 127 KEGG pathways. Once again, the biosynthesis of flavonoids and flavonols was significant (Figure 3B), and DEGs in the metabolism of valine, leucine, isoleucine, fatty acid and α-linolenic acid metabolism were identified. α-Linolenic acid metabolism is an important raw material for JA synthesis, which involves multiple JA pathway-related genes. Therefore, our results indicate that ABA affected genes are involved in the JA synthesis pathway.

### 2.4. Changes in JA and ABA Hormone Content and Gene Expression after BPH Infestation

Hormone signals interact with each other in plants, and the determination of hormone content can help us further study the interaction mechanism of plant hormones under the infestation of brown planthoppers. Changes in ABA and JA content were measured in rice plants treated with BPH, BPH + JA and BPH + ABA, respectively. One aim of this study was to elucidate interactions between JA and ABA. ABA content was two-fold higher in the BPH + JA treatment as compared with the control (BPH only). The BPH + ABA treatment was 60-fold higher compared with BPH (Figure 4A). The JA content was not significantly different from the control (BPH only) in the BPH + ABA treatment. Exogenous JA (see BPH + JA treatment) increased JA content ~68-fold relative to the control (Figure 4B).

The JA biosynthesis and signal transduction pathways are shown in Figure 5A. After treatment with BPH + ABA, several genes in the JA synthesis pathway were up-regulated, including *OsAOS2*, *OsOPR*, *OsACOX* and *OsACAA1*. Interestingly, *OsLOX1* was down-regulated, which may impact the final concentration of JA (Table 3). Rice homologues in the JAZ family (*OsJAZ11*, jasmonate ZIM-domain protein) also showed elevated expression in the BPH + ABA treatment (Table 3). The results indicate that ABA can induce JA synthesis and increase the expression of JAZ proteins.

Exogenous JA increased the expression of *OsNCED1*9, which encoded 9-cis-epoxycarotenoid dioxygenase, an important gene in the ABA synthesis pathway (Figure 5B). JA increased the increased expression of *OsPP2C*, which encoded the protein phosphatase 2C clade A protein; however, *OsPYL9* (pyrabactin resistance-like ABA receptor) and *OsSAPK1* (stress-activated protein kinase) showed significant reductions in expression in the BPH + JA treatment (Table 4). These results indicate that the JA treatment can impact ABA synthesis and ABA-mediated signal transduction in plants.

### 2.5. Expression of Rice Transcription Factors in Response to Hormone Treatments

TFs can function in various cellular processes, including development, cell cycle regulation and the organismal response to environmental stress. Our transcriptome data identified 232 transcriptional factors that were regulated by ABA and/or JA. After ABA application, 68 and 92 TFs were up- and down-regulated, respectively. In response to the JA application, 67 and 62 TFs were up- and down-regulated, respectively. We recorded 57 genes encoding TFs that could be assigned to 21 different families; 23 TF genes were up-regulated, and 34 were down-regulated (Figure 6A,B). Genes belonging to the bZIP, MYB and WRKY families were up-regulated in response to ABA and JA; these TFs function in plant resistance to external stresses. These data suggest that both the interaction between hormones and TF gene expression improve rice resistance.

### 2.6. Quantitative RT-PCR

The following six genes were selected for RT-qPCR analysis: the TF *OsMYC2*, *OsABA8OX2* (ABA 8′-hydroxylase), *OsIAA23* (auxin-responsive gene), TFs *OsZIP23* and *OsPP2C8* and *OsKSL4* (ent-kaurene synthase-like protein). The expression levels of these six genes were compared by RNA-seq and qPCR in rice treated with BPH + ABA (Figure 7A) and BPH + JA (Figure 7B). There was a correlation between RNA-seq expression and the qPCR data, which confirmed the reliability of the transcriptome data.

## 3. Discussion

Multiple studies have been conducted on hormone-induced plant resistance to biotic and abiotic stress. For example, mutations in *aba2-1* improved the survival rate of herbivores in Arabidopsis [23], whereas *NaHER1* (herbivore elicitor-regulated protein 1) inhibited ABA metabolism and reduced insect resistance in tobacco [17]. ABA-induced deposition of callose, a cell wall polysaccharide, reduce the ingress of pathogens and viruses in host plants piercing-sucking insects such as BPH are known to induce callose deposition in the phloem and stimulate the production of trypsin inhibitors, thereby reducing the feeding of BPH [14,24]. The findings reported herein further support the contention that plants are able to adjust their defense reaction in response to herbivores.

Early signaling events play a vital role in the induction of herbivore-associated molecular responses. For example, aphid infestation in *Medicago truncatula* up-regulated the ABA signaling pathway, which triggered stomatal closure and reduced leaf transpiration [25]. Hillwig et al. (2016) [26] found that 4-methoxyindole-3-ylmethylgluconate, which had an anti-insect effect in *Arabidopsis* ABA deletion mutant *aba1-1*, was more abundant than that in wild type (WT), indicating the accumulation of ABA signals to reduce the accumulation of resistant compounds to be beneficial to the aphids. ABA could also ally with TFs responses to BPH; bZIP played an important role in abiotic stress. Many TFs regulate ABA-mediated plant signal transduction, for which the bZIP family plays an essential role [27]. Transgenic plants overproducing OsbZIP TFs exhibited strong sensitivity to ABA, improved drought and salt tolerance and up-regulated levels of many stress resistance-related genes [28,29,30,31]. MYB TFs regulate plant growth and development and also play vital roles in abiotic and biotic stress [32,33,34]. WRKY TFs play a central role in both plant growth and the plant stress response, especially in response to BPH [35,36]. In summary, our transcriptome results indicate that the exogenous application of ABA enhanced rice resistance to abiotic stress by up-regulating the expression of multiple TF-encoding genes, including members of the *OsbZIP*, *MYB* and *WRKY* families.

Plants rely on the JA signaling pathway to produce direct and indirect defense responses to insects [37]. BPH induced the expression of JA-mediated pathogenesis-related genes *NbPR3* and *NbPR4*, which resulted in induced resistance [38]. Similarly, exogenous MeJA induced resistance to BPH in rice [39]. JA stimulates changes in plant volatiles that are similar to those caused by herbivorous insects, thus producing an indirect defense response. For example, exogenous JA induced the production of rice volatiles and improved parasitism of BPH by the rice wasp, *Anagrus nilaparvatae* [40]. Exogenous JA increased the release of volatile α-terpene and β-ocimene from tomato, which inhibited the whitefly *Bemisia tabaci* [41]. In the present study, BPH-activated genes involved ‘α-linolenic acid metabolism’, and this resulted in increased levels of JA. Both JA and ethylene function together in the plant defense response; for example, genes encoding the EIN3-binding F-box protein (*OsEBF1*) and ethylene-insensitive like 1 (*OsEIL1*) regulated BPH infestation in the ethylene signaling pathway. *OsEBF*1 mediated the degradation of *OsEIL1* through the ubiquitin pathway, while *OsEIL1* directly regulated the JA biosynthesis gene, *OsLOX9* [42]. JA and ET pathways can respond to piercing-sucking insects synergistically. Taken together, JA and its crosstalk with other hormones in rice are crucial for plant defense against herbivory.

Hormone signaling pathways in plants have multiple roles and may function synergistically or antagonistically. Many plants rely on JA signaling for direct or indirect defense against insects [37]. JA induces secondary metabolic pathways and can perturb plant hormone signaling networks in complex ways [43]. For example, the transcription factor MYC2 in the JA signaling pathway functions in the crosstalk between ABA and MeJA [44] and activates ABA signal transduction [45]. Further studies showed that ABA and JA signals in Arabidopsis interact through PYL6 and MYC2, and *pyl6* mutants showed increased sensitivity to JA and ABA [43]. The gene encoding Arabidopsis ABA-insensitive 4 (*ABI4*) was necessary for ABA-dependent growth regulation and JA-dependent signal transduction. Low levels of ABA stimulated ABA- and JA-dependent signaling pathways, which mediated ABI4 regulation of plant growth [46]. ABA and JA signals also modulate plant defense against chewing insects. Dinh et al. (2013) [16] suggested that oral secretions of the tobacco diamondback moth (*Manduca sexta*) induced the ABA signal transduction pathway, which enhanced the JA-mediated defense response. Exogenous ABA also enhanced the expression of MYC2 in the Arabidopsis JA signaling pathway. The ABA mutant *aba2-1* blocks the expression of *VSP1*, which is regulated by MYC2, indicating that ABA regulates plant-induced insect resistance and activates JA-mediated defense responses [17].

Crosstalk between ABA and JA signal transduction pathways also functions to optimize the plant response to drought. Water-limiting conditions significantly up-regulated the ABA signaling pathway of alfalfa and increased JA-dependent defenses, which prolonged the time required for *Acyrthosiphon*
*pisum* to reach the phloem [47]. Nguyen et al. (2016) [48] found that drought and *Spodoptera exigua* feeding increased the levels of ABA and JA in *Solanum dulcamara*, making drought-stressed plants more resistant to insects than fully-watered plants.

Our findings further illustrate the interactions between JA and ABA in modulating BPH resistance on rice and reveal key genes involved in defense. These data, taken together, showed that JA and ABA had synergistic interrelationships in the regulation of the plant response to biotic and abiotic stress.

## 4. Materials and Methods

### 4.1. Plant Materials

Seeds of rice cultivar ZH11 were soaked for 2–3 d and germinated in an RX intelligent artificial climate incubator at 26 ± 2 °C, 70–80% RH and a 16/8 h light/dark photoperiod. After 15 d, rice seedlings were transplanted into small buckets in each barrel.

### 4.2. ABA, JA and BPH Treatments

At the tillering stage, individual plants were sprayed with an ABA (50 μmol/L) or JA (50 μmol/L) solution containing 0.5% Tween; control plants were sprayed with 0.5% Tween. Treatments consisted of BPH, ABA + BPH, or JA + BPH and contained three replicates. After spraying for 12 h, the outermost leaf sheath of rice was removed and frozen in liquid nitrogen. In BPH treatments, 3rd instar BPH nymphs (*n* = 30) were allowed to feed on rice; after 12 h, the outermost leaf sheath was removed and frozen in liquid nitrogen.

### 4.3. RNA Library Construction, Sequencing and Quality Control

The Takara Minibest Plant RNA Extraction kit was used for RNA extraction, and the purity, concentration and integrity of RNA samples were analyzed by NanoDrop, Qubit 2.0 and Agilent 2100, respectively. The mRNA was enriched with oligo (dT) beads and randomly interrupted adding fragmentation buffer; the first cDNA chain was synthesized with mRNA as a template, and the second cDNA chain was synthesized by adding buffer, dNTPs, RNase H and DNA polymerase I. The purified cDNA was purified with AMPure XP beads, and a poly-A tail was added and sequenced; the fragment size was selected using AMPure XP beads and enriched by PCR to obtain the cDNA library. The quality and yield of the library were then analyzed by Qubit 2.0, Agilent 2100 pairs and qRT-PCR.

### 4.4. RNA Sequencing

The Illumina sequencing platform was used to analyze rice transcriptome data (Nanjing Jisihuiyuan Biotechnology Company, Nanjing, China). Sequence alignments were carried out between the clean data generated in this study and the rice reference genome, and mapped data were obtained. DEGs were analyzed according to the expression level in the sample group, and the DEGs were evaluated by GO functional annotation through Gene Ontology annotations [49] and KEGG pathway enrichment analysis.

### 4.5. Quantification of Phytohormones

UPLC-ESI-MS/MS analysis was used to detect plant hormones in samples with an API 5500 triple quadrupole tandem mass spectrometry detection system (AB Sciex, Massachusetts, MA, USA) that was equipped with an electrospray ionization (ESI) ion source and the Analyst 1.6.2 workstation. Using ESI, analytes were evaluated using the multi-reaction detection mode by negative ion scanning, which greatly improves sensitivity. Mass spectrometry parameters and methods such as the declustering potential and capillary electrophoresis were used to quickly screen and determine the ion pairs of target compounds under optimal conditions. The optimized mass spectrometry conditions were as follows: air curtain gas, 30; collision gas, 8; ion spray voltage, −3000 V; temperature, 500 °C; ion source gas, 1:35 and gas, 2:45.

An Ultra-Performance Liquid Chromatography System (Waters) was used for hormone detection. Compounds were separated on an Agilent InfinityLab Poroshell 120 EC-C18 column (100 × 3 mm, 2.7 μm), and the injection volume was 3 μL. Solvents included mobile phase A (0.1% formic acid in water) and mobile phase B (0.1% formic acid in acetonitrile).

### 4.6. Identification of DEGs and Verification by qRT-PCR

DEGs were detected using DESeq2 software; log_2_ fold change ≥1 and false discovery rates (FDR) <0.05 were used as screening criteria. Fold change represents the ratio of expression among groups, and FDRs were obtained by correcting the significance *p* values. Six DEGs were selected for real-time quantitative PCR, and the accuracy of transcriptome sequencing was verified. The HiScript^®^ Q RT SuperMix for qPCR (+gDNA wiper) (Vazyme) was used for reverse transcription, and Primer Premier 5.0 was used for primer design. Quantitative PCR was performed using the ChamQ^TM^ SYBR^®^ Color qPCR Master Mix Kit (Vazyme). The mixed solution was added to 96-well plates, and the reaction mixture contained the following: primers of the target gene, 0.8 μL; 2× ChamQ SYBR Color qPCR Master Mix, 10 μL; and template, 2 μL. cDNA from rice sheaths was used as a template, *ACTIN I* was selected as the internal reference, and each sample was tested thrice. The relative expression of each gene was calculated by the 2^−ΔΔCt^ method [50] (Livak et al. 2001).

### 4.7. Statistical Analysis

FPKM values were calculated for gene expression analysis. DEGs were defined as those with *p*-adjust ≤0.05 and a default difference multiple = 2. The SPSS v. 11.0 statistical software package was used to analyze the data. Univariate or multifactorial ANOVA and multiple comparisons were performed by the PLSD method. Descriptive statistics were expressed as means + standard error, and the significance level was set to *p =* 0.05.

## 5. Conclusions

In summary, the transcriptome profiling in this study revealed dynamic transcriptional and metabolic changes in rice in response to BPH infestation. BPH feeding induced rapid and precise defense responses that were involved in many different processes. Future studies will focus on the roles and underlying molecular mechanisms of signaling events and TFs. This study also provides genetic clues that could be used to improve rice defense against BPH.

## Figures and Tables

**Figure 1 ijms-23-06319-f001:**
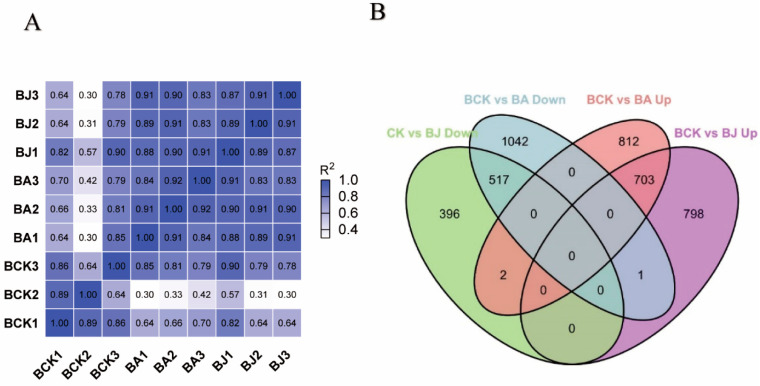
Pearson correlation between samples and DEGs in BPH, BPH + ABA and BPH + JA treatments. (**A**) Pearson correlation between nine samples. (**B**) Up-regulated and down-regulated DEGs in rice in response to BPH infestation where BPH was the control. Abbreviations: BCK, CK + BPH as control; BA, BPH + ABA; and BJ, BPH + JA.

**Figure 2 ijms-23-06319-f002:**
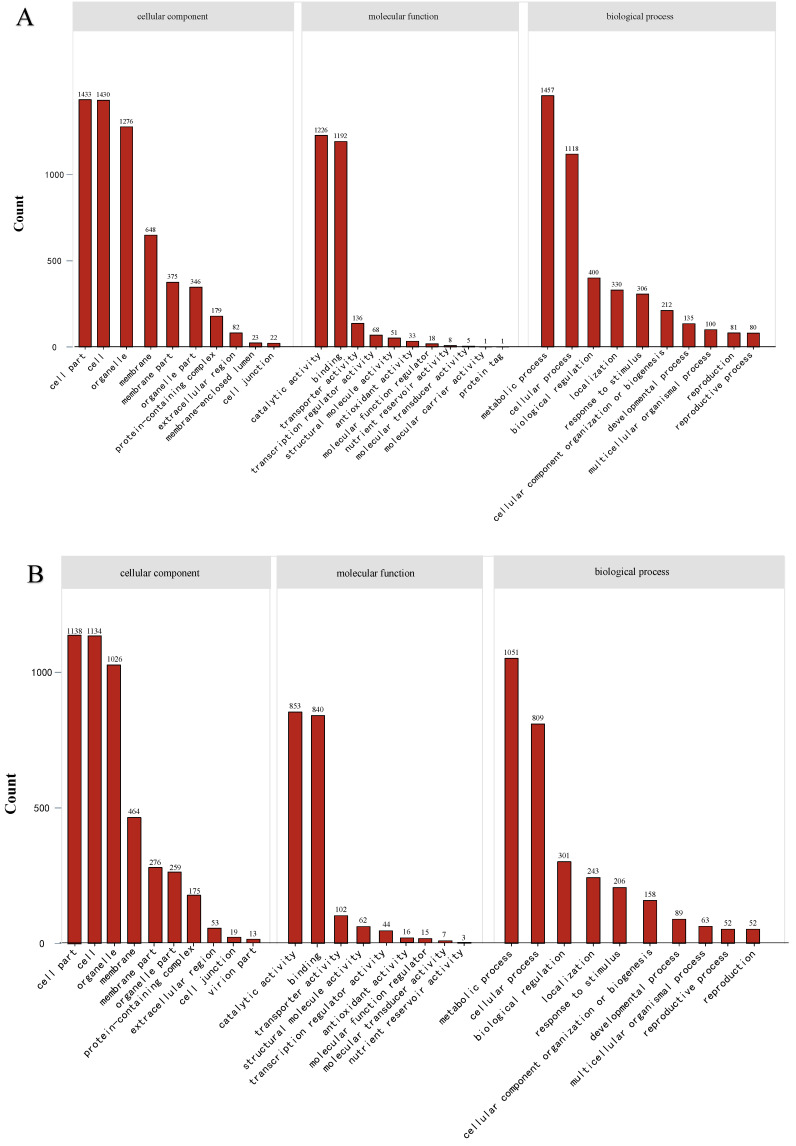
The top 30 enriched gene ontology terms in rice DEGs infested with BPHs and treated with exogenous ABA and JA. Panels (**A**,**B**) show results from BPH + ABA and BPH + JA treatments, respectively. KOG function gene classification of the DEGs. (**A**,**B**) The top 30 terms enriched gene ontology. (**C**) Distribution of DEGs in the KOG database.

**Figure 3 ijms-23-06319-f003:**
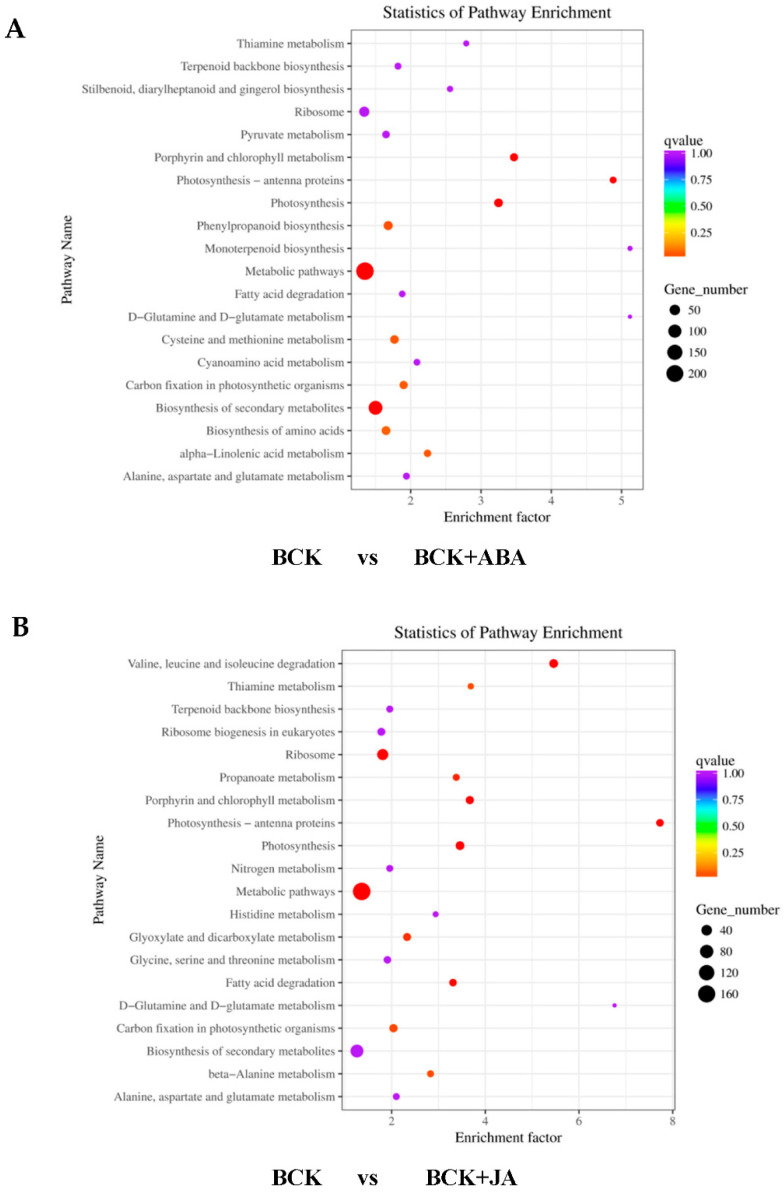
KEGG pathway enrichment scatter plots of DEGs. Pathway names are on the left. The x-axes show enrichment factors, and larger enrichment factors indicate greater significance in expression levels. Q-values are indicated on the right. Dot size represents the number of genes, and colors represent different Q values, which are *p*-values corrected by multiple hypothesis tests. (**A**) KEGG pathway of DEGs under BCK vs. BCK + ABA. (**B**) KEGG pathway of DEGs under BCK vs. BCK + JA. Abbreviations: BCK, CK + BPH.

**Figure 4 ijms-23-06319-f004:**
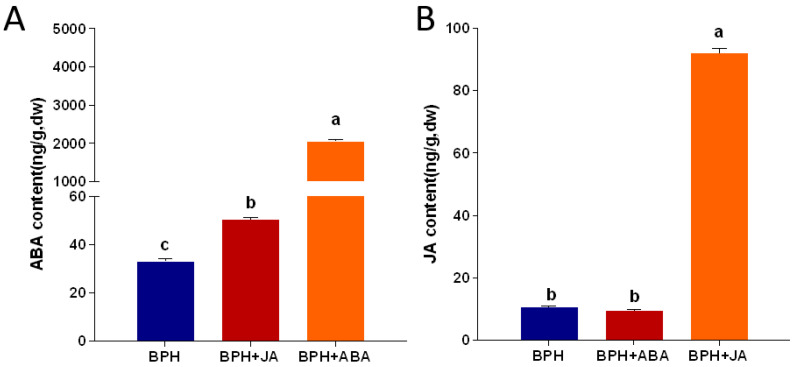
Growth hormone content in rice sheaths treated with BPH (control), BPH + JA and BPH + ABA. (**A**) ABA content after treatment with BPH, BPH + JA and BPH + ABA. (**B**) JA content after treatment with BPH, BPH + ABA and BPH + JA. Data represent means + SE. Columns labeled with different letters indicated significant differences at *p* < 0.05 (*PLSD* test).

**Figure 5 ijms-23-06319-f005:**
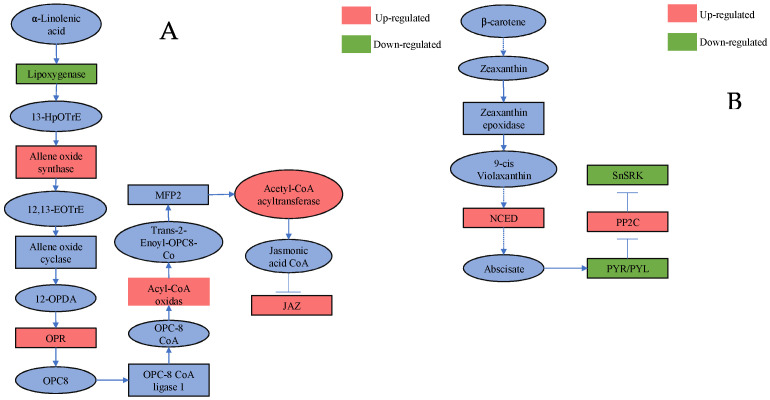
Schematic diagram showing genes involved in JA (**A**) and ABA (**B**) biosynthesis and signal transduction. Genes surrounded by red shading are up-regulated, whereas green shading represents down-regulated genes.

**Figure 6 ijms-23-06319-f006:**
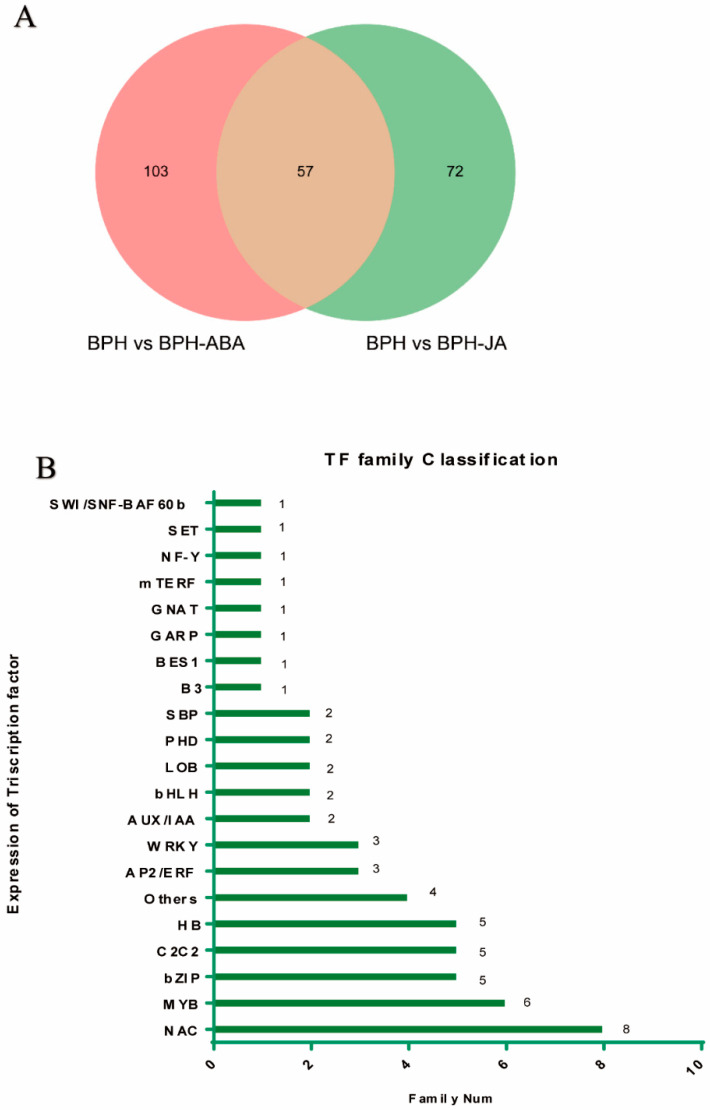
Classification of differentially-expressed transcription factors in response to hormone treatment. (**A**) Venn diagram of DEGs identified in BPH vs. BPH + ABA and BPH vs. BPH + JA treatment groups. (**B**) Number of DEGs in 21 TF families. These DEGs were differentially expressed in the BPH vs. BPH + ABA and BPH vs. BPH + JA treatments. The ordinate shows the expression levels of the 21 TF families, and the number of DEGs representing a family is shown on the abscissa.

**Figure 7 ijms-23-06319-f007:**
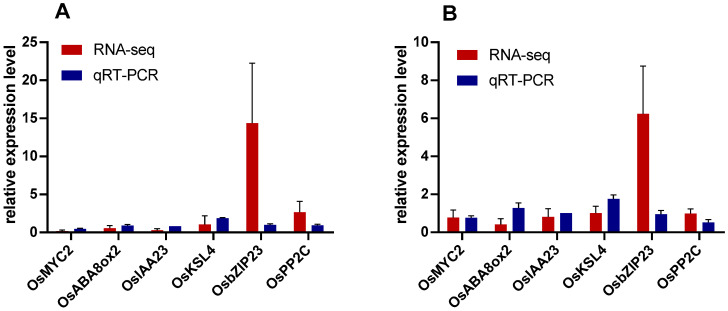
Comparison expression results obtained by RNA-seq and RT-qPCR for six DEGs. (**A**) The expression of six genes in rice treated with BPH + ABA; (**B**) expression level of six genes in rice treated with BPH + JA. Relative expression levels were measured using the 2^−ΔΔt^ method. Data are expressed as means + SE. Different letters indicated significant differences (*PLSD* test, *p* < 0.05).

**Table 1 ijms-23-06319-t001:** Summary of RNA-seq results.

Samples	Read Number	Base Number	% ≥Q30	Reads Aligned	Exonic
BPH1	29004617	8701385100	92.74	51063904 (88.03%)	35,168,954 (82.64%)
BPH2	23242201	6972660300	92.24	37905421 (81.54%)	22,739,057 (79.34%)
BPH3	26550424	7965127200	92.54	45041229 (84.82%)	33,151,024 (84.38%)
BPH + ABA1	26551481	7965444300	92.38	48669904 (91.65%)	36,079,719 (84.7%)
BPH + ABA2	29957564	8987269200	94.73	53778358 (89.76%)	36,055,653 (83.67%)
BPH + ABA3	36308246	10892473800	95.25	60138179 (82.82%)	37,540,070 (81.7%)
BPH + JA1	27950890	8385267000	95.37	51810937 (92.68%)	38,656,560 (86.07%)
BPH + JA2	28574794	8572438200	93.63	54632427 (95.60%)	42,975,066 (86.27%)
BPH + JA3	32635686	9790705800	95.31	61075150 (93.57%)	44,054,381 (85.55%)

**Table 2 ijms-23-06319-t002:** DEGs unique to ABA and JA treatments.

Treatment	All DEGs	Up-Regulated DEGs	Down-Regulated DEGs
BPH vs. BPH + ABA	3491	1795	1696
BPH vs. BPH + JA	2727	1674	1053

**Table 3 ijms-23-06319-t003:** JA pathway genes showing differential expression to BPH + ABA treatment of rice.

	Pathway ID	Gene Name	Functional Annotation	*p*-Value	log_2_FC
LOC_Os02g10120	K00454	*OsLOX1*	Lipoxygenase	8.38 × 10^−12^	−3.58132
LOC_Os03g12500	K01723	*OsAOS2*	Oxidoreductase activity	2.94 × 10^−9^	3.099447
LOC_Os08g35740	K05894	*OsOPR*	12-oxophytodienoic acid reductase	7.45 × 10^−5^	2.655998
LOC_Os05g07090	K00232	*OsACOX1*	Acyl-CoA oxidase	6.23 × 10^−5^	2.050977
LOC_Os06g24704	K00232	*OsACOX3*	Acyl-CoA oxidase	7.72 × 10^−9^	1.708204
LOC_Os02g57260	K07513	*OsACAA1*	Acetyl-CoA acyltransferase 1	7.89 × 10^−10^	1.422267
LOC_Os03g08320	K13464	*OsJAZ11*	Jasmonate ZIM domain-containing protein	5.65 × 10^−3^	1.686453
LOC_Os03g28940	K13464	*OsJAZ6*	Jasmonate ZIM domain-containing protein	3.51 × 10^−4^	1.574457
LOC_Os07g42370	K13464	*OsJAZ7*	Jasmonate ZIM domain-containing protein	3.37 × 10^−5^	1.789341
LOC_Os09g26780	K13464	*OsJAZ8*	Jasmonate ZIM domain-containing protein	2.33 × 10^−3^	2.0051
LOC_Os10g25290	K13464	*OsJAZ12*	Jasmonate ZIM domain-containing protein	2.49 × 10^−7^	2.292083
FC, fold change					

**Table 4 ijms-23-06319-t004:** ABA pathway genes showing differential expression to BPH + JA treatment of rice.

	Pathway ID	Gene Name	Functional Annotation	*p*-Value	log_2_FC
LOC_Os06g36670	K14496	*OsPYL9; OsPYL/RCAR9*	Abscisic acid receptor, PYR/PYL family	7.97 × 10^−4^	−1.63928
LOC_Os02g47510	K09840	*OsNCED1*	9-cis-epoxycarotenoid dioxygenase	3.01 × 10^−4^	1.875411
LOC_Os01g46760	K14497	*PP2C*	Protein serine/threonine phosphatase activity	4.4 × 10^−3^	1.37321
LOC_Os03g27280	K14498	*OsSAPK1*	Serine/threonine-protein kinase SRK2	4.40 × 10^−5^	−1.33978
FC, fold change					

## Data Availability

Not applicable.

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
