# Peer review of "Transcriptome Analysis Reveals Crosstalk between the Abscisic Acid and Jasmonic Acid Signaling Pathways in Rice-Mediated Defense against Nilaparvata lugens"

_ijms, 2022, doi:10.3390/ijms23116319_

Round 1

Reviewer 1 Report

Authors improved manuscript based on reviewer's comments.

Author Response

Dear reviewer,

Thank you for your comments and suggestions.

We appreciate that!

Hope you have a nice day!

Reviewer 2 Report

I checked your manuscript and described comments below.

I think this paper is a good analysis of the abscisic acid and jasmonic acid signaling pathways in rice-mediated defense against Nilaparvata lugens.

I think “The DEGs identified 23 in this study provide vital insights into the synergism between ABA and 24 JA and further contribute to the mechanistic basis of rice resistance to BPH” is especially important.

There is one problem. I think it's bad to post a manuscript that came back from proofreading. You should post with the text color black.

There are some minor problems with this paper below.

1.       The characters in the figures are too small to read.

2.       It is better to add the contents of differentially expressed genes (DEGs) as supplementary materials..

Author Response

Dear Reviewer,

We (all the authors) appreciate that our manuscript is under minor revision. And thanks to the reviewer for your hard-working. We revise the final version and make some changes:

  1. On page 1, we add the author’s information.
  2. According to the reviewer’s suggestion, we enlarge figure 2 so that makes it easy to read. And we add the excel for contents of differentially expressed genes as supplementary materials.
  3. We change the format of Fig. 5 which makes it clear to read.
  4. We add the full stop at the end of the references and revise the format of reference number 21.
  5. We have checked that all references are relevant to the contents of the manuscript.
  6. We use MS word and mark up the revision using the “Track changes”.

This manuscript is a resubmission of an earlier submission. The following is a list of the peer review reports and author responses from that submission.

Round 1

Reviewer 1 Report

The manuscript “Transcriptome analysis revealed cross-talk between abscisic acid and jasmonic acid signaling pathway in rice resistance against Nilaparvata lugens” mentioned for intraction between JA and ABA under BPH attacking based on transcriptome analysis. The research topic is important for understanding plant response by exogenous application. However, I have following comments and questions.

  1. General
    According to comparisons between CK+BPH and BPH+ABA and between CK+BPH and BPH+JA, you can just identify the gene response to ABA and JA based on my understanding. Because all of treatments has same conditions of BPH attacking and you cannot mention that the genes with DEGs related to resistance against BPH. At least, you should additionally compare between CK and ABA and between CK and JA and you can estimate which genes response to BPH attacking under exogenous applications, JA and ABA.
  2. Abstract and Introduction
    Is there any studies for enhancing BPH resistance on rice plants by exogenous application, JA and ABA? You mentioned that “Exogenous application of abscisic acid (ABA) and jasmonic acid (JA) increased rice resistance against BPH.” in abstract. If so, you must add explanation and reference in Introduction.
  3. Materials and methods, Page 2 Line 78

How about BPH resistance levels of ZH11? There is no information of resistance genes or susceptible to BPH on ZH11.

  1. Materials and methods, Page 2 Line 83 to 89

You must explain the reason (or reference) of concentrations of JA and ABA.

  1. Result

Authors did not show result of BPH resistance level of ZH11 on each treatment, CK+BPH, BPH+ABA and BPH+JA. You must add the result for resistance of ZH11 under three treatments. If not, there is no evidence of enhancing resistance by exogenous application of JA and ABA with 50 μmol/L ABA and 50 μmol/L  JA solutions. Therefore, the relationship between exogenous application and BPH resistance on ZH11 is unclear based on current data.

Author Response

We would like to express our thanks to you for your work on our submission and for the revision comments. We would also like to thank you for your valuable comments on the manuscript.

Our  responses  to each comments are in the attachment. please see the attachment.

Reviewer 2 Report

In their manuscript, Li et al. show that ABA and JA influence  transcriptional response of rice to a pest brown planthopper (BPH). The authors performed RNAseq analysis upon BPH and hormone treatment as well as determined JA and ABA in the treated plants. The study provides some new insights into how JA ana ABA can modulate rice resitance to BPH. The idea is interesting and the authors obtained relevant RNA-seq data that may be important for further applicative research. However, the results are not clearly presented and interpreted, and I foun that the manuscript suffers from a number of mistakes. Below are specific comments.

1) It is not clear whether ABA /JA treatment impact only transcriptional response of rice. Could Authors observe any changes in the phenotype of BPH treated plants after ABA or JA treatment ?

2) Some data presented in the figures are left without any comment or interpretation in the main text, for example Fig. 1B - what may be a significance of the observed overlaps ?

3) It is not clear what are the data presented on the Fig. 7. What was the control in this experiment? 

4) Fig. 5, can Authors provide additional qPCR results supporting expression changes of JA and ABA biosynthetic genes ?

5) what was the basis of selection of the 6 genes for qPCR?

6) Authors should avoid general statements like "some studies had shown". Instead, particular works should be cited. 

7) Figure legends are often not sufficient and lack relevant data. For example, in the Fig. 6B it is unknown which condition was used for the analysis of TF families. JA? ABA? both? In addition, Fig. 6, A and B panels are not described 

8) it is not clear whether hormone tratment was performed before, after, or simultaneously with BPH ?

9) It is hard to understand from the discussion what are the Authors` major conclusions about the influence of ABA and JA on BPH treatment.

Other comments:

there are many language mistakes throughout the text

some graphs and descriptions are hardly visible : for example, Fig. 2

the reason why the Authors focus on particular genes or TF groups are not clear in the abstract and introduction

Author Response

(The authors gave the same response as above.)

Round 2

Reviewer 1 Report

The manuscript “Transcriptome analysis revealed cross-talk between abscisic acid and jasmonic acid signaling pathway in rice resistance against Nilaparvata lugens” mentioned for intraction between JA and ABA under BPH attacking based on transcriptome analysis. The authors revised manuscript according to reviewer comments.

Reviewer 2 Report

The Authors have made some corrections in the manuscript. However, some remarks have not been addressed: 

The remark #3 concerned a control used for normalisation as only data from BPH + ABA/JA are shown in the Figure 7, but the relative level of each gene is different. Therefore, it is still unknown what data are actually presented there. They might repreent the differences between BPH + ABA/JA vs BPH (or BCK ? this is inconsistent throughout the text) but one can only guess.

Remark #4 concerned Fig.5, while the Authors refer to Fig. 7 and do not provide suggested qPCR data.

There are many typing errors, and incorrect English expressions. Language corrections are necessary for this manuscript.